

**Measurement report: characterization and sources of the ambient secondary organic carbon in a**
**Chinese megacity over five years from 2016 to 2020**
Meng Wang[1], Yusen Duan[2], Wei Xu[3], Qiyuan Wang[4], Zhuozhi Zhang[1], Qi Yuan[1], Xinwei Li[1], Shuwen Han[1], Haijie
Tong[1], Juntao Huo[2], Jia Chen[2], Shan Gao[5], Zhongbiao Wu[6], Long Cui[4], Yu Huang[4], Guangli Xiu[7], Junji Cao[4, 8],
Qingyan Fu[2, *], Shun-cheng Lee[1, *]
[1]Department of Civil and Environmental Engineering, The Hong Kong Polytechnic University, Hung Hom, Hong
Kong
[2]Shanghai Environmental Monitoring Center, Shanghai, China
[3]School of Physics, Ryan Institute's Centre for Climate & Air Pollution Studies, and Marine Renewable Energy Ireland,
National University of Ireland Galway, University Road, Galway, H91 CF50, Ireland
[4]State Key Laboratory of Loess and Quaternary Geology, Institute of Earth Environment, Chinese Academy of
Sciences, Xi'an 710061, China
[5]Zhejiang Tianlan Environmental Protection Technology Co., Ltd., Hangzhou 311202, China
[6]Department of Environmental Engineering, Zhejiang University, 866 Yuhangtang Road, Hangzhou, 310058, China
[7]State Environmental Protection Key Laboratory of Environmental Risk Assessment and Control on Chemical Process,
School of Resources and Environmental Engineering, East China University of Science and Technology, Shanghai
200237, China
[8]Key Laboratory of Middle Atmosphere and Global Environment Observation, Institute of Atmospheric Physics,
Chinese Academy of Sciences, Beijing 100029, China
*Correspondence to*: shun-cheng.lee@polyu.edu.hk (S.C. Lee) and qingyanf@sheemc.cn (Q.Y. Fu).
**Abstract**
To investigate impact factors and source area of secondary organic aerosols in the Yangtze River Delta (YRD) region,
a world-class urban agglomeration in China, long-term measurements of organic carbon (OC) and elementary carbon
(EC) in particulate matter of less than 2.5 μm ($PM_{2.5}$) with hourly time resolution were conducted at a regional site in
Shanghai from 2016 to 2020. Based on the five-year measurements, the interannual, monthly, seasonal, and diurnal
variations in OC and EC, as well as OC subtypes, i.e., secondary OC (SOC) and primary OC (POC), apportioned by
the novel statistical model of the minimum $R^2$ method, and the formation pathways of SOC, are presented. By
examining the relationship between SOC and temperature, as well as relative humidity (RH), we show that SOC
formation is greatly enhanced at high temperatures (>30 °C), while it is inversely correlated with RH. In particular,
we show that the photochemical formation of SOC is the major formation pathway even in winter when solar radiation
was supposedly less intense than in summer, which is different from that in north China plain where aqueous phase
chemistry is found to be an important SOC formation pathway. Moreover, increased SOC concentrations are also
found to be associated with high wind speed (>5 m s$^{-1}$) in winter, which is increased by 29.1% (2.62 μg m$^{-3}$) when
compared to that during lower winds, suggesting regional sources of SOC in winter. By analyzing the potential source
regions using the concentration weighted trajectory (CWT), the geographic regions of SOC are found to be mainly
associated with transport from outside Shanghai (SOC > 3.5 μg m$^{-3}$) including central and southern Anhui, Zhejiang,
and Fujian. The results from this study provide critical information about the long-term trend of carbonaceous aerosol,





in particular SOC, in one of the largest megacities in the world and are helpful to develop pollution control measures
from a long-term planning perspective.

**Keywords:** PM$_{2.5}$; Carbonaceous aerosols; Secondary organic carbon (SOC); Long-term observation; Concentration-
weighted trajectory (CWT)

## 1 Introduction

Carbonaceous aerosols account for 20-90% of the submicron aerosol mass (Jimenez et al., 2009; Kroll et al., 2011).
It affects the physical and chemical properties of the atmosphere, including radiative forcing, hygroscopicity, and
toxicity (Hopke, 1991; Pope and Dockery, 2006; Bond et al., 2013). Carbonaceous components are classified
experimentally into three fractions: elemental carbon (EC), carbonate carbon (CC) and organic carbon (OC) (Turpin
et al., 2000). EC is a primary pollutant that can be directly emitted from fossil fuel combustion and biomass burning
(Cao et al., 2003; Galindo et al., 2019). Carbonate carbon is mainly in natural mineral dust and building/demolition
dust and exists in the coarse fraction (Chow and Watson, 2002; Chang et al., 2017), while, OC is composed of hundreds
of organic compounds, forming a complex mixture with different chemical and physical properties and accounting for
a major fraction of the carbonaceous aerosol (Chatterjee et al., 2021). OC can either be emitted directly from e.g.,
combustion processes, vehicular exhaust and cooking and is termed as primary OC (POC). It can also be formed in
the atmosphere by gas-to-particle oxidation reactions, terms as secondary organic carbon (SOC) (Salvador et al., 2021;
Hallquist et al., 2009). Carbonaceous aerosols are among the major constituents of atmospheric aerosols and their
quantification is necessary for understanding the role of aerosols in issues varying from the regional visibility
degradation to health effects and global climate change (Wu et al., 2012; Mauderly and Chow, 2008).
Since carbonaceous aerosols are indispensable for probing atmospheric aging processes of organic aerosols and
formulating effective emission control policies, there have been a number of studies in China. In the 1980s, Dod et al.
(1986) first published a study on carbonaceous aerosol in Beijing in three seasons (i.e., spring, summer and winter),
demonstrating that ambient carbonaceous aerosols (soot in the study) were derived principally from coal combustion,
especially in winter. With the development of analytical techniques and in-depth research on carbonaceous aerosols,
a number of studies on the carbonaceous aerosols have emerged in many Chinese cities, such as Beijing (He et al.,
2001; Zhang et al., 2007; Ji et al., 2019), Xi'an (Cao et al., 2005; Han et al., 2009; Shen et al., 2014), Chengdu (Wang
et al., 2013; Tao et al., 2013), Shanghai (Cao et al., 2013; Zhu et al., 2015; Li et al., 2019), Guangzhou (Cao et al.,
2004; Ho et al., 2014; Wang et al., 2016), and Hong Kong (Lee et al., 2006; Ho et al., 2002; Ho et al., 2019). Cao et
al. (2007) conducted the first nationwide simultaneous measurements of carbonaceous aerosols in 14 cities in China
in the winter and summer of 2003, revealing the seasonal and regional sources of carbonaceous aerosols across China.
However, long-term (e.g., 5 years) analysis of carbonaceous aerosols in the megacities is currently lacking, limiting
our understanding of the trend evolution of carbonaceous aerosols and the ability to evaluate the effectiveness of air
quality policies such as "Action Plan on Prevention and Control of Air Pollution" (Zhang et al., 2019).
Shanghai is one of the megacities with the most rapid economic and social development in the Yangtze River Delta
(YRD), China (Lin et al., 2014). Along with rapid economic growth and urbanization, the consequent degradation of
air quality has been recognized (Fu et al., 2008; Wang et al., 2015). Hence, the Chinese government unveiled its 5-
year "Action Plan on Prevention and Control of Air Pollution" in 2013, a comprehensive guideline that calls for
nationwide improvements in air quality by 2017, aiming to cut PM$_{2.5}$ levels by 20% in the regions of YRD (MEP,


2013). Over the past decade, extensive studies have been launched to investigate the impact of carbonaceous aerosol
on air pollution in Shanghai. The $PM_{2.5}$ reduction targets in Shanghai have been met to date (Zhang et al., 2019).
However, it is not well understood how the $PM_{2.5}$ components, particularly carbonaceous aerosol, were evolving over
recent years, with different components likely demonstrating distinct temporal evolution. For example, while Shanghai
has witnessed a decrease in EC concentration, from an annual average value of 2.81 µg m$^{-3}$ in 2010 to 2.11 µg m$^{-3}$ in
2014, it also saw a small increase in OC concentration, from an annual average value of 7.09 µg m$^{-3}$ in 2010 to 7.83
µg m$^{-3}$ in 2014 (Chang et al., 2017). To grasp more complete information on the variation, evolution and sources of
the carbonaceous aerosol, especially in the post- "Action Plan on Prevention and Control of Air Pollution" era,
continuous and highly time-resolved measurements of carbonaceous aerosol over multiple years are necessary but are
currently lacking.

In this study, we conducted a long-term field campaign at a regional site in the YRD region from 2016 to 2020.

Hourly time-resolved OC and EC in $PM_{2.5}$ were measured in a supersite in Shanghai. The secondary organic carbon
(SOC) was estimated by the minimum $R^2$ method (MRS) (Wu and Yu, 2016). The characteristics of carbonous aerosol
pollution and their seasonal and diurnal variations are discussed. Furthermore, we explored the meteorological factor
effects on carbonaceous aerosol concentrations in different levels of $PM_{2.5}$. To attain a better understanding of the
temporal variations of SOC in different $PM_{2.5}$ levels and source areas, we identified the main source areas of SOC by
employing backward trajectory clusters and the Concentration weighted trajectory (CWT) model based on the Hybrid
Single Particle Lagrangian Integrated Trajectory (HYSPLIT) analysis. The purpose of this study is to improve the
understanding of the variation and sources of SOC in the $PM_{2.5}$ fraction. The long-term data presented in this study
provides critical information that can evaluate the effectiveness of the current air pollution control policies and are
informing to develop future pollution control measures.
**2 Experiment and method**
**2.1 Observation site**

The sampling site is in Qingpu District of western Shanghai, named Dianshan Lake (DSL) supersite

(31.09°N,120.98°E, ~ 15 m above ground) (Fig. S1). It is ~7 km east of Dianshan Lake, ~ 50 km from downtown
Shanghai, situated at the intersection area of Jiangsu, Shanghai and Zhejiang. It is in a suburban area in the Yangtze
River Delta, surrounded by farmland and vegetated lands. There are two highways (G318 and G50, ~1 km to the site)
but no large industries near the sampling site. DSL station is a supersite maintained by the Shanghai Environmental
Monitoring Center and the monitoring data is incorporated into the national regional air automatic monitoring network
of China. The site is considered suitable to investigate the regional air quality and transport of air pollutants in YRD
region (Jia et al., 2020). In this study, a five-year intensive campaign was conducted at DSL site from January 2016
to December 2020, of which the observations were suspended from July to September 2019 (5% of the data) due to
site maintenance.
**2.2 Instruments and measurements**

$PM_{2.5}$ mass concentrations were determined automatically by a tapered-element oscillating microbalance monitor

(TEOM, Thermo FH62C-14, USA). The sampling flow rate of the TEOM was 16.7 L min$^{-1}$. The uncertainty of the
hourly measurement is ± 1.50 mg m$^{-3}$, and the detection limit is 0.1 µg m$^{-3}$. In the study, the $PM_{2.5}$ concentration was



converted to hourly means. $O_3$ and $NO_2$ were measured by an online analyzer (Model O342M, Environmental S.A,
FRA; model 42i, Thermo Environmental Instruments, USA).
Organic carbon (OC) and elemental carbon (EC) were measured online by a Sunset Semi-Continuous Carbon
Analyzer (Sunset Laboratory, Forest Grove, Oregon, USA) using the thermal-optical transmittance method at a flow
rate of 8 L min$^{-1}$. This instrument can provide hourly time-resolved OC and EC analyses. The detection limit of OC
and EC are 0.2 and 0.04 μg m$^{-3}$, respectively.
The meteorological parameters including ambient temperature (T), relative humidity (RH), wind speed (WS),
and wind direction (WD) were obtained at the sampling site using the Visala (WXT520, Vaisala Ltd., Finland)
automatic weather station at hourly time resolution. The uncertainty of ambient temperature, RH, WS and WD are ±
0.1 °C, ± 3%, ± 0.3 m s$^{-1}$ and 3°, respectively. The data is collected every minute and converted to hourly means.
**2.3 Estimation of secondary organic carbon (SOC) by minimum $R^2$ (MRS) method**
Since EC is a tracer for primary POC from combustion sources, EC-tracer method has been widely used for
separating POC and SOC (Cao et al., 2007). In this study, an innovative EC-tracer method was used to estimate SOC
named the minimum correlation coefficient (MRS) method (Wu and Yu, 2016). The concentrations of SOC were
estimated as follows
$$POC = EC \times (OC/EC)_{primary} \qquad (1)$$
$$SOC = OC - POC \qquad (2)$$
where OC and EC are the concentrations measured in the sample, $(OC/EC)_{primary}$ is an estimate of the primary OC/EC
ratio through calculating a hypothetical set of $(OC/EC)_{primary}$. The hypothetical $(OC/EC)_{primary}$ that generates the
minimum correlation coefficient ($R^2$) SOC values was determined by seeking the minimum $R^2$ between SOC and EC.
This method may result in negative SOC concentrations for those periods when the estimated $(OC/EC)_{primary}$ value
was higher than the measured OC/EC ratio. Although these data increase the uncertainty of the method, we assumed
these points were free of SOC formation. Since the relative contributions of different primary emission sources would
vary from month to month (Table S1), we calculated $(OC/EC)_{primary}$ according to this method for each month from
2016 to 2020 in Shanghai (Fig. S2-S6).
**2.4 Back trajectory and concentration-weighted trajectory (CWT) model**
To determine the influences of regional transport on SOC at Shanghai, we calculated 72 h air mass back trajectory
of the central location at 500 m above the ground level. The trajectories were calculated with the NOAA Hybrid Single
Particle Lagrangian Integrated Trajectory (HYSPLIT4.0) model (Draxler and Rolph, 2003). The meteorological data
were from the Global Data Assimilation System (GDAS). The model was run eight times per day at starting times of
00:00, 03:00, 6:00, 9:00, 12:00, 15:00, 18:00, and 21:00 local time (LT), respectively. The relative parameter settings
in the model had also been used in the literature (Wang et al., 2018; Lin et al., 2019).
The concentration-weighted trajectory (CWT) approach was used to investigate the potential transport of pollution
(Fleming et al., 2012) on the interface of ZeFir (Petit et al., 2017). For the CWT calculations, the entire geographic
region covered by the 3-day backward trajectories was separated into 7920 grid cells of 0.5° latitude × 0.5° longitude.
Each grid cell was assigned a residence-time-weighted concentration obtained by the hourly averaged SOC
concentration associated with the trajectories that crossed that grid cell (Hsu et al., 2003). The CWT is defined as:



$$C_{ij} = \frac{\sum_{l=1}^{M} C_l \tau_{ijl}}{\sum_{l=1}^{M} \tau_{ijl}} \qquad (3)$$

where $C_{ij}$ is the average weighted concentration in the grid cell ($ij$th); $C_l$ is the measured SOC concentration on the
arrival of trajectory $l$; $\tau_{ijl}$ is the number of trajectory endpoints in the $ij$th grid cell by trajectory $l$; and M is the total
number of trajectories.
**3 Results and discussion**
**3.1 Temporal variations of carbonaceous aerosol**
**3.1.1 Interannual variations**
Summary statistics for carbonaceous aerosol concentrations (EC, OC, POC, and SOC), as well as total $PM_{2.5}$, from
1 January 2016 to 31 December 2020 are presented in Table 1. During the entire observation period, the EC
concentration ranged from 0.01-11.6 µg m$^{-3}$, and the five-year average concentration was $1.28 \pm 0.95$ µg m$^{-3}$. Annually,
the EC concentration measured at Dianshan Lake has essentially decreased year by year in parallel over the five years.
The average concentration of EC was highest in 2016, with an annual average of $1.50 \pm 1.17$ µg m$^{-3}$, while the average
concentration of EC in 2020 was the lowest ($1.00 \pm 0.6\ 4$ µg m$^{-3}$). Therefore, compared to EC in 216, the annual EC
concentration reduced by ~ 50% in 2020.
Different from EC, the average concentration of OC was the highest in 2017 (average $6.32 \pm 3.52$ µg m$^{-3}$). Since
2018, the average concentration of OC has decreased year by year with the lowest annual level of $4.99 \pm 2.93$ µg m$^{-3}$
found in 2020. It is worth noting that although the average concentration of OC in 2016 was lower than that in 2017,
the maximum concentration of OC in 2016 was 1.41-1.61 times that of other years.
OC subtypes of POC and SOC were apportioned using the novel MRS statistical model (See the method section).
Different from the trend observed for EC, the apportioned POC increased year to year from 2016 to 2019, reaching
the maximum value in 2019 ($3.76 \pm 2.55$ µg m$^{-3}$). However, it dropped sharply in 2020. Therefore, the POC/EC ratio
was changing for different years, and using a fixed POC/EC value over multiple years might bias the POC as well as
SOC. In contrast, the changing trend of SOC was consistent with OC, which was the maximum in 2017 (average SOC
concentration $2.98 \pm 2.25$ µg m$^{-3}$). In the next three years, the annual average concentration of SOC decreased on a
year-to-year basis, reaching the lowest in 2020 ($1.53 \pm 1.35$ µg m$^{-3}$).
**3.1.2 Seasonal and monthly variations**
The seasonal variations of carbonaceous aerosol concentrations are illustrated in Fig. 1. The season-wise average
concentrations of EC ranged from 0.92 (summer of 2019) to 1.90 µg m$^{-3}$ (winter of 2016), while OC ranged from 4.35
(summer of 2012) to 7.83 µg m$^{-3}$ (winter of 2016). For EC, OC, as well as OC subtypes (POC and SOC), similar
seasonal variations are observed with generally higher average carbonaceous aerosol concentrations in autumn and
winter and lower levels in spring and summer, except for a slightly higher concentration of EC in the summer of 2017
due to the boost in intensive pollution episodes (indicated by a significantly higher value in 95th percentile than that
in other years). Higher concentrations of EC were observed in winter for the other four years, which could be caused
by the stagnation of the atmosphere and the stronger influence of regional transport during wintertime (Chen et al.,
2017).





There is a consistent pattern for the seasonal variations of POC concentrations, the concentration levels of POC in
spring, summer, and fall were generally lower than that in the winter, reflecting generally locally-dominated POC
emissions in Dianshan Lake. In particular, POC concentrations in winter were 5.40, 3.88, 4.10, 4.13, and 3.97 $\mu g\ m^{-3}$,
and 2.04, 1.31, 1.40, 1.38, and 1.41 times higher than those in the summer for 2016, 2017, 2018, 2019, and 2020,
respectively.
SOC concentrations were estimated to be ranging from 0.13 ~10.70 $\mu g\ m^{-3}$ (spring), 1.04 ~ 19.41 $\mu g\ m^{-3}$ (summer),
0.02 ~ 25.37 $\mu g\ m^{-3}$ (autumn), and 0.03 ~ 37.14 $\mu g\ m^{-3}$ (winter) over the five years. Comparatively, there was no clear
trend in the seasonal changes of SOC over the five years, which can be explained by their complexity in terms of the
sources and formation processes. Indeed, in contrast to POC, SOC is the mixed product of the aging of the primary
emissions and secondary formation from precursor gases, which could vary significantly in different seasons. For
instance, strong solar radiation tends to facilitate photochemical reactions and thus enhance the formation of volatile
organic compounds (VOCs) to organic aerosols in summer (Tuet et al., 2017), while the increased anthropogenic
emissions (e.g., biomass burning and coal burning emissions) will also lead to a significant increase in SOC during
the harvest period and heating season (Zhang et al., 2013; Wang et al., 2020).
Monthly, the average mass concentrations of carbonaceous aerosols show relatively large variations in this study
(Table S1), with the average value ranging from 0.56 (October 2017) to 2.22 (December 2017) $\mu g\ m^{-3}$ for EC, while
it ranged from 3.39 (October 2017) to 9.00 (December 2017) $\mu g\ m^{-3}$ for OC. The month of December presented the
highest EC and OC average concentration (EC: 1.81 ± 1.36 $\mu g\ m^{-3}$; OC: 7.27 ± 5.03 $\mu g\ m^{-3}$) throughout the study
period. The lowest month for carbonaceous aerosols concentration was in August (EC: 0.94 ± 0.52 $\mu g\ m^{-3}$; OC: 4.47
± 2.56 $\mu g\ m^{-3}$). These are consistent with the previous study in Shanghai from 2010 to 2014 (Chang et al., 2017).
Table S1 also shows the monthly mean POC and SOC concentrations at our study site for the whole 5-year period.
POC shows similar variations to OC, with higher average concentrations in the cold season (from November to
February next year) and lower ones in the warm season (from April to October). The highest average POC
concentration was 4.97 ± 3.97 $\mu g\ m^{-3}$ (December), and the lowest POC average concentration was 2.23 ± 1.34 $\mu g\ m^{-3}$
(August). In contrast, the SOC average concentration was the highest in July (3.43 ± 3.12 $\mu g\ m^{-3}$), which accounted
for 58.1% on average of OC in the same month.
**3.1.3 Weekend–weekday pattern and diurnal variations**
Fig. 2 shows diurnal patterns of carbonaceous aerosols during weekdays and weekends in four seasons, as well as
over the entire study period. Consistently, EC shows a distinctive diurnal pattern for different seasons or the whole
period (Fig. 2), which is characterized by two peaks occurring in the morning (around 08:00 local time) and during
the evening (around 20:00 local time), corresponding well with the morning and evening rush hours, coupled by
shallow mixing layer heights. It is worth noting that the peak of EC during the morning peak is higher than the evening
peak, and the difference between the two peaks is the largest in winter and the smallest in spring.
Different from EC, the daily variation of OC does not show a consistent pattern (Fig. 2). OC shows a peak at around
noon in spring and summer, while the peak time in autumn and winter is advanced to about 10:00 in the morning. The
peak appearing near noon can reflect the contribution of photochemical reaction to OC. In particular, the apportioned
SOC shows increased concentrations at a similar time. This phenomenon is especially obvious in spring and summer,
while no clear change in the concentration of SOC is found in winter. Additionally, in autumn and winter, OC also
shows a peak at 22:00, which is partly due to the primary emission as evident by the simultaneous increase in POC,
while such increase is absent in SOC (Fig. 2).





In terms of weekdays and weekends variation, the average concentration of EC during weekends in spring and
autumn is higher than that of weekdays (spring $EC_{weekdays}$= 1.25 µg m$^{-3}$; $EC_{weekends}$=1.30 µg m$^{-3}$, autumn
$EC_{weekdays}$=1.19 µg m$^{-3}$; $EC_{weekends}$= 1.24 µg m$^{-3}$), while the difference between weekday and weekends is small in
summer (both at ~1.10 µg m$^{-3}$). Only the winter $EC_{weekdays}$ (1.56 µg m$^{-3}$) is higher than $EC_{weekends}$ (1.46 µg m$^{-3}$). The
weekday and weekend variation observed at this site is different when compared to previous studies. Specifically,
according to a previous literature report (Chang et al., 2017), the observational data from 2010 to 2014 showed that
the concentration of EC on working days was greater than that on weekends because the traffic volume was
significantly higher on weekdays than on weekends, consistent with the location of the sampling site that is near
national highways (2 km away). However, Shanghai has officially implemented a traffic restriction system in 2016.
In this study, the sampling site is located near tourist attractions and is not in the traffic restricted area of Shanghai,
which is near the national expressway entering and leaving Shanghai (the straight-line distance is no more than two
kilometers). It is speculated that the heavy traffic flow due to the attraction of the nearby tourist sites during spring
and autumn weekends may lead to high EC emissions.
POC and SOC show different weekly patterns. Specifically, the concentration on working days in winter is higher
than that on weekends, while the SOC weekend in spring is slightly higher than that of weekdays, and the weekdays
of other seasons are higher than weekends (Fig. 2). This indicates that there is no significant decline in anthropogenic
activity on the weekends compared to weekdays. Enhanced anthropogenic emissions could be caused by no limit on
driving vehicles based on license plates on weekends. Human activities increase near the sampling site, leading to
increased VOC emissions and more SOC generation. Below we discuss more the sources of SOC and the impact of
meteorological parameters on its formation.
**3.2 Insights into the formation pathways of SOC**
**3.2.1 Relationship between SOC and temperature**
Examining the relationship between SOC vs. meteorological parameters (e.g., temperature, RH, and wind speed)
could provide more information on the formation and transformation of ambient SOC. Fig. S7 shows the statistics on
the concentration distribution of SOC in different temperature bins. Specifically, the mean value of SOC concentration
was 2.42 µg m$^{-3}$ (T < 0 °C), 2.32 µg m$^{-3}$ (0 °C < T < 10 °C), 2.06 µg m$^{-3}$ (10 °C < T < 20 °C), 1.98 µg m$^{-3}$ (20 °C <
T < 30 °C), and 3.82 µg m$^{-3}$ (T > 30 °C) during the study period. Therefore, while the concentration of SOC does not
show a linear increase with the increase in temperature, at T > 30 ℃, the SOC is significantly higher than in other
groups. We further use the T-test in different temperature groups and find that their difference is statistical significantly
(Fig. S8).
To investigate the temperature impacts on the formation of secondary organic aerosols, we divided the dataset
into four groups based on $PM_{2.5}$ concentrations for all seasons (Fig. 3). The clean periods were defined for $PM_{2.5}$
concentration < 15 µg m$^{-3}$, the transition periods were defined for 15 µg m$^{-3}$ < $PM_{2.5}$ < 35 µg m$^{-3}$, the less polluted
days were defined for 35 µg m$^{-3}$ < $PM_{2.5}$ < 100 µg m$^{-3}$, and the severe haze periods were defined for conditions with
$PM_{2.5}$ > 100 µg m$^{-3}$. The definition of clean and haze periods is based on the national primary ambient air quality
standards for annual and daily mean $PM_{2.5}$ concentrations (i.e., 15 and 35 µg m$^{-3}$, respectively). Below we show that
the promotion of SOC at high temperatures (>30 ºC) is held true for pollution levels in all seasons.
Specifically, during the clean period in spring, the SOC concentration in Dianshan Lake in spring showed a trend
of first decreasing and then increasing with temperature. When 10 °C < T < 20 °C, the average SOC concentration





was the lowest (1.13 µg m$^{-3}$). However, when T > 30 °C, the highest SOC concentration (3.41 µg m$^{-3}$) was more than
doubled. Under the transition and mild pollution conditions, the change of SOC also showed a minimum value at
10 °C < T < 20 °C, but the concentration at this low point increased with the intensification of pollution. On heavy
pollution days, when the temperature is less than 30 °C, the temperature has no obvious promoting effect on the
generation of SOC. Similarly, during the clean period in summer, the effect of temperature increase on SOC was not
significant. However, under transition and pollution conditions (including light pollution and severe pollution), the
average concentration of SOC will increase significantly with the increase in temperature. Especially during periods
of severe pollution, the SOC concentration increased from 1.93 µg m$^{-3}$ (10 °C < T < 20 °C) to 9.30 µg m$^{-3}$ (T > 30 °C).
In autumn, except for the clean days when the mean SOC concentration was the highest (1.46 µg m$^{-3}$) at 10 °C < T <
20 °C, the average SOC concentration in the pollution period increases with the increase of temperature for other
periods. In comparison, winter SOC (The bottom panel in Fig. 3) is most significantly affected by temperature during
periods of severe pollution. During the severe pollution period, when 20 °C < T < 30 °C, the average concentration of
SOC was higher than the average concentration of SOC under all conditions in other seasons, on average, reaching
10.0 µg m$^{-3}$.
In order to further verify the effect of temperature on the SOC concentration under various pollution conditions,
we conducted the Pearson correlation test between different temperature intervals and SOC concentration in each
period (Fig. S9). The results show that during the clean period, the Pearson correlation coefficient between temperature
and SOC concentration is only 0.31 (T < 0 °C), indicating that the effect of temperature on the average concentration
of SOC is not significant during the clean period. The highest values of Pearson's correlation coefficients appeared at
T > 30 °C under transitional and lightly polluted conditions, but none of them exceeded 0.5. However, during the
period of heavy pollution at T > 30 °C, the Pearson correlation coefficient between SOC and the temperature increased
to 0.62 (Fig. S9), demonstrating a more significant role of temperature in driving SOC formation during the heavy
pollution periods.
**3.2.2 Relationship between SOC and RH**
Fig. 4 shows the diurnal variations of SOC concentrations and RH in four different PM$_{2.5}$ groups. In general, RH is at
its highest in the early morning and lowest between 13:00 and 15:00 noon. Most of the peaks of SOC during the
pollution period of each season appear at the lowest RH value at noon, while such a pattern is not observed during
clean periods. Specifically, during clean periods in spring, the daily average of RH is 70.4%, and the daily average
concentration of SOC is 1.26 µg m$^{-3}$. The peak of SOC appeared at 1:00 am when the relative humidity reached 77.4%,
and then the relative humidity continues to increase gradually, reaching the highest value of the day (RH: 87.0%) at
7:00. However, the concentration of SOC does not change significantly, all around 1.30 µg m$^{-3}$. This indicates that on
clean days, SOC is not significantly affected by photochemistry. In contrast, during more polluted periods in spring,
SOC shows an increased concentration (> 2.10 µg m$^{-3}$) at 15:00, which is due to photochemical oxidation which
overcomes the dilution effects caused by the increased planetary boundary layer in the afternoon. In summer, the
change of RH in different PM$_{2.5}$ ranges is not obvious with a mean RH of 78%, but the difference in SOC concentration
is significant. The daily average concentration of SOC in severe pollution is roughly 5 times that of clean days. In the
clean periods of summer, the nighttime peak of SOC is 1.61 µg m$^{-3}$, which is larger than the daytime peak of 1.52 µg
m$^{-3}$. With the intensification of the pollution degree, the difference between the peak daytime SOC and the peak
nighttime SOC gradually increased in summer with low RH associated with high SOC in the afternoon.
During heavy pollution in winter, RH does not change significantly between 0:00 and 6:00, and the nighttime





peak of SOC appears at 1:00 (SOC: 3.75 μg m$^{-3}$). During the day, the RH gradually decreased to the lowest value of
50.4% at 14:00. At the same time, the concentration of SOC increases significantly and remains at a high concentration
level from 9:00 to 16:00, suggesting the photochemical formation of SOC is still very efficient and important even in
winter when solar radiation was supposedly less intense than in summer.

**3.2.3 Photochemical formation of SOC**

The oxidant $O_x$ ($O_x = O_3 + NO_2$) is usually used as a proxy to indicate the atmospheric oxidizing capacity associated
with photochemical reactions (Wang et al., 2017). The daily $O_x$ minimum occurred in the morning followed by a sharp
increase to a peak in the afternoon in all seasons (Fig. S10). Similarly, SOC also a large increase in the afternoon in
all seasons, with peak concentrations in the range of 2.40-3.00 μg m$^{-3}$ (Fig. S10). The concurrent increase in SOC and
$O_x$ in the afternoon suggests photochemical formation was a dominant formation pathway for SOC even in winter.
This is different from the formation pathways of SOC in north China, where aqueous phase chemistry is often reported
to be the major formation pathway of SOC in winter (Lin et al., 2020; An et al., 2019; Sun et al., 2015; Chen et al.,

2019).

The positive relationship between SOC and $O_x$ was well presented in four different $PM_{2.5}$ bins in different seasons
(Fig. 5). In spring, SOC was positively correlated with $O_x$ with the concentrations of SOC during the haze periods ~
1.8−3.2 times higher than those during the clean periods. The SOC in summer and fall showed a similar trend with
higher levels of $O_x$ significantly associated with the increased SOC concentrations. The average concentration of SOC
reached its highest during the severe haze period in summer and autumn with an average SOC of > 6.00 μg m$^{-3}$ when
$O_x$ was > 200 μg m$^{-3}$. For the $PM_{2.5}$ bin of > 100 μg m$^{-3}$ in winter, the concentration of SOC showed a significant
increase (>6.00 μg m$^{-3}$; mean value) from <4 .00 μg m$^{-3}$ when $O_x$ increased to >200 μg m$^{-3}$ from < 50 μg m$^{-3}$. In
contrast, the increase in SOC for other $PM_{2.5}$ bins was less significant in winter, due to the generally low $O_x$ for $PM_{2.5}$
of < 100 μg m$^{-3}$ (Fig. 5).

**3.2.4 Relationship between SOC and wind speed/direction**

Wind speed is an important factor controlling the concentrations of carbonaceous aerosols. In this study, EC and
POC concentrations show evident WS dependence, with higher concentrations in association with lower wind speeds
(Fig. S11). This is consistent with the general pattern that pollution episodes are likely to occur under lower wind
speeds (WS < 1 m s$^{-1}$) (Ren, 2018). At the same time, the relationship between the concentration of carbonaceous
aerosols and wind speed can also reflect that its main contribution comes from local emissions or regional transmission.
In particular, in spring, summer, and autumn, the concentration of carbonaceous aerosols decreased with the increase
in wind speed, indicating that in these seasons, local emissions at low wind speeds are the main contribution of
carbonaceous aerosols. It was worth noting that in winter, on the one hand, the concentration of carbonaceous aerosol
under each wind speed gradient is higher than that of other seasons. On the other hand, when the wind speed is higher
than 4.5 m s$^{-1}$, the concentration of carbonaceous aerosol is also increased. Specifically, the concentration of EC
increased by 12.4%, while POC increased by 11.7%, indicating the contribution of the transport in winter to
carbonaceous aerosols.
In contrast, SOC is affected differently by wind speed. The dependence of SOC concentrations on wind speed is
shown in Fig. 6a. In spring, the concentration of SOC is about 2 μg m$^{-3}$, and the concentration gradient of SOC
increases slightly with the increase in wind speed. When the wind speed is greater than 1.5 m s$^{-1}$ and less than 2 m s$^{-1}$,



the concentration of SOC reaches the highest value of 2.15 μg m⁻³. When the wind speed is less than 0.5 m s⁻¹, the
SOC concentration is 1.37 μg m⁻³, and when the wind speed is greater than 5 m s⁻¹, the SOC concentration is 1.78 μg
m⁻³, with an increase of 29.8%. In summer, the concentration of SOC decreases with the wind speed gradient. When
the wind speed is 2~2.5 m/s, the SOC concentration is the highest (2.79 μg m⁻³). When the wind speed is greater than
5 m/s, the SOC concentration is at its lowest (1.40 μg m⁻³). In autumn, the SOC does not appear to change significantly
(~2 μg m⁻³) when the wind speed gradient gradually increased. In winter, when the wind speed is less than 4.5m s⁻¹,
the SOC concentration is about 2 μg m⁻³ (mean value is 2.03 μg m⁻³). When the wind speed is greater than 4.5 m s⁻¹,
the SOC concentration increases to 2.45 μg m⁻³. It is worth noting that when the wind speed is greater than 5 m s⁻¹,
the concentration of SOC increases by 29.1% (2.62 μg m⁻³), reaching the highest average concentration of SOC under
different wind speed gradients in winter. This shows that the main contribution of Shanghai SOC in winter comes
from regional transmission.

The seasonal bivariate polar plots of SOC concentrations for 2016 – 2020 were shown in Fig. 6b. The high

concentration load of SOC near the sampling site in all seasons mainly occurs in the southwest direction and under
the condition of low wind speed (WS less than 4 m/s). The concentration distributions of SOC (Fig. 6b) and OC were
similar in spring (Fig. S12c), and the highest concentration area appeared in the southwest region. The distribution
and loading of SOC with a high concentration in summer (SOC > 4 μg m⁻³) is closer to the sampling point (dense
distribution in WS around 2 m s⁻¹), further proving the previous conclusion that the main contribution of SOC in
summer from a local build. The relationship between SOC and wind direction remains unchanged in autumn. However,
the high SOC loading area is still located southwest of the sampling point, but the concentration is significantly lower
than that in summer and autumn which is closer to the sampling point. In addition, in the southeast direction, the area
with wind speed greater than 6 m s⁻¹ has a high loading area of SOC, and it is speculated that this part of SOC may
be transmitted from the area near the east coast.
**3.3 Analysis of potential source regions of SOC**

The CWT results demonstrate the spatial distributions of SOC in the form of the SOC weighted 72-h backward

trajectories (Fig. 7). The CWT results are generally consistent with the corresponding polar plots as shown in Section
3.2.4. Specifically, the potential source areas with high CWT values for SOC were located in the surroundings of
Shanghai. In spring, SOC mainly comes from North China and the middle and lower reaches of the Yangtze River,
specifically from central Anhui, southern Jiangsu, central and northern Zhejiang, and northern Fujian; in summer, the
high SOC values in southern Shanghai (SOC > 3.5 μg m⁻³) mainly come from central and southern Anhui, Zhejiang,
Fujian; there are also great contributions from offshore (the northern South China Sea and East China Sea). The SOC
in Shanghai in autumn mainly comes from the northern and central regions of Zhejiang. Southern Jiangsu is the main
source of SOC in Shanghai in winter, followed by northern Zhejiang; on the other hand, the northern long-distance
transmission from the North China Plain further extends to Inner Mongolia, Mongolia and the Russian border.

We further analyzed the potential sources of Shanghai SOC under different PM concentrations (Fig. 8). The

concentrations of SOC during the clean period were all lower than 2 μg m⁻³, and there were three main source
pathways, namely, the northern of the North China Plain, Inner Mongolia and eastern Mongolia; the Yellow Sea and
the Korean Peninsula; Zhejiang Province and northern Fujian Province. The source area of SOC during the transition
period was further expanded, and the concentration of SOC in the main area was between 2 and 3 μg m⁻³, which was
basically consistent with the source area coverage during the cleaning period. It is worth noting that a high SOC
loading appeared in the coastal area of Fujian, presumably related to secondary aerosols transported by oceanic air



masses. During the transition periods ($35 < PM < 100$ µg m$^{-3}$), the source area of SOC expanded, and the area with
SOC concentration greater than 3.5 µg m$^{-3}$ became clearer, mainly concentrated in Zhejiang, Fujian and offshore areas.
The main transmission areas are eastern Inner Mongolia, Hebei Province, and the North China Plain, all the way
southward to Anhui, connecting with the source areas of Zhejiang Province and Fujian Province. During high pollution
periods, the areas with SOC concentrations higher than 4 µg m$^{-3}$ were mainly concentrated in two areas, one was from
southeastern Mongolia in the north, through eastern Inner Mongolia and Hebei Province, through Shandong and
Jiangsu and finally to Shanghai; the other source area was in the south of Shanghai Zhejiang Province, Fujian Province,
and eastern Jiangxi Province. Overall, the main potential source areas of Shanghai SOC are the Yangtze River Delta,
North China Plain, northern China, Inner Mongolia, and eastern China provinces and offshore areas, such as Zhejiang
Province, Fujian Province, and the South China Sea.
**4 Conclusions**
In this study, the hourly mass concentration of OC and EC in PM$_{2.5}$ were continuously measured from 1 January
2016 to 31 December 2020 at a supersite in Shanghai. OC subtypes of POC and SOC were estimated by the novel
MRS method. Based on the five-year measurements, the interannual, monthly, seasonal, and diurnal variations in OC
and EC, as well as OC subtypes are presented. By examining the relationship between SOC and meteorological
parameters (e.g., temperature, RH, and wind speed), as well as O$_x$, the sources, formation and transformation
mechanisms of ambient SOC are revealed. We show that SOC formation is greatly enhanced at high temperatures
($>30$ °C), while it is inversely correlated with RH. In particular, we show that the photochemical formation of SOC is
still very efficient and is the major formation pathway even in winter when solar radiation was supposedly less intense
than in summer. High EC and POC concentrations are found to be associated with low wind speeds, which is consistent
with their primary nature from local emission. Moreover, increased SOC concentrations are also found to be associated
with high wind speed ($>5$ m s$^{-1}$) in winter, which is increased by 29.1% (2.62 µg m$^{-3}$) when compared to that during
lower winds, suggesting regional sources of SOC in winter. By analyzing the potential source regions using the CWT
algorithm, the geographic regions of SOC are found to be mainly associated with transport from outside Shanghai
(SOC $> 3.5$ µg m$^{-3}$) including central and southern Anhui, Zhejiang, and Fujian.
**Data availability**
The data presented in this study are available at the Zenodo data archive https://doi.org/10.5281/zenodo.6473085
(Wang et al., 2022).
**Supplement.**
The supplement related to this article is available online at:



**Declaration of competing interest**

The authors declare that they have no known competing financial interests or personal relationships that could have appeared to influence the work reported in this paper.

**Credit authorship contribution statement**

Meng Wang: Conceptualization, Methodology, Validation, Formal analysis, Writing- original draft.

Yusen Duan: Methodology, Formal analysis.

Wei Xu: Validation, Formal analysis.

Qiyuan Wang: Conceptualization, Writing, Review and Editing.

Zhuozhi Zhang: Formal analysis, Writing, Review and Editing.

Qi Yuan: Formal analysis, Methodology.

Xinwei Li: Investigation, Methodology.

Shuwen Han: Validation, Investigation.

Haijie Tong: Writing, Review and Editing.

Juntao Huo: Investigation, Methodology, and Validation.

Jia Chen: Investigation, Methodology, and Validation.

Shan Gao: Methodology, Validation, Formal analysis.

Zhongbiao Wu: Review and editing, Funding acquisition.

Long Cui: Formal analysis, Investigation.

Yu Huang: Writing, Review and Editing.

Junji Cao: Writing, Review and Editing.

Qingyan Fu: Writing, Review editing, Funding acquisition, and Resources.

Shun-cheng Lee: Writing, Review editing, Funding acquisition, and Supervision.

**Acknowledgements**

This work was supported by the Environment and Conservation Fund - Environmental Research, Technology Demonstration and Conference Projects (ECF 63/2019), the RGC Theme-based Research Scheme (T24-504/17-N), the RGC Theme-based Research Scheme (T31-603/21-N), the Key Research and Development Projects of Shanghai Science and Technology Commission (20dz1204000), the Key Research and Development Projects of Shanghai Science and Technology Commission (9DZ1205000) and Hangzhou Qianjiang Distinguished Experts Project. We also thank the contribution of the State Ecology and Environment Scientific Observation and Research Station for the Yangtze River Delta at Dianshan Lake.



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

**Table 1** Averages, medians, and associated standard deviations for the OC, EC, POC, SOC and PM$_{2.5}$ concentrations (in µg m$^{-3}$) from Jan. 2016 to Dec. 2020.

| | | 2016 | 2017 | 2018 | 2019 | 2020 | Whole study |
|---|---|---|---|---|---|---|---|
| **EC** | Average | 1.50±1.17 | 1.23±0.88 | 1.31±0.88 | 1.31±0.89 | 1.00±0.64 | 1.28±0.95 |
| | Median | 1.18 | 1.01 | 1.04 | 1.07 | 0.82 | 1.01 |
| | Range | 0.07~11.57 | 0.01~6.27 | 0.01~9.07 | 0.08~6.85 | 0.14~5.46 | 0.01~11.57 |
| **OC** | Average | 6.03±4.01 | 6.32±3.52 | 5.79±3.58 | 5.40±3.16 | 4.99±2.93 | 5.75±3.53 |
| | Median | 4.93 | 5.61 | 4.87 | 4.53 | 4.15 | 4.83 |
| | Range | 0.77~41.85 | 0.41~29.49 | 0.78~29.77 | 0.78~25.96 | 0.57~26.40 | 0.41~41.85 |
| **POC** | Average | 3.48±3.23 | 3.34±2.40 | 3.61±2.67 | 3.76±2.55 | 3.45±2.27 | 3.52±2.67 |
| | Median | 2.48 | 2.72 | 2.81 | 3.06 | 2.83 | 2.77 |
| | Range | 0.13~37.14 | 0.02~19.41 | 0.03~22.55 | 0.19~20.71 | 0.42~17.05 | 0.02~37.14 |
| **SOC** | Average | 2.56±1.94 | 2.98±2.25 | 2.17±1.75 | 1.64±1.20 | 1.53±1.35 | 2.24±1.87 |
| | Median | 2.10 | 2.38 | 1.71 | 1.41 | 1.20 | 1.76 |
| | Range | 0.01~18.13 | 0.01~25.79 | 0.01~19.87 | 0.01~18.84 | 0.01~14.87 | 0.01~25.79 |
| **TCA** | Average | 7.53±5.06 | 7.55±4.29 | 7.10±4.38 | 6.72±3.98 | 5.98±3.50 | 7.03±4.36 |
| | Median | 6.10 | 6.66 | 5.98 | 5.64 | 4.99 | 5.88 |
| | Range | 0.94~53.42 | 0.44~31.91 | 1.07~34.65 | 0.96~31.74 | 0.83~30.20 | 0.44~53.42 |
| **PM$_{2.5}$** | Average | 53.0±36.16 | 44.9±31.48 | 45.16±34.22 | 48.18±32.82 | 40.14±28.96 | 46.50±33.25 |
| | Median | 43.0 | 37.0 | 35.0 | 38.0 | 31.0 | 37.0 |



| Range | 1.0~219.0 | 1.0~299.0 | 1.0~258.0 | 1.0~220.0 | 1.0~236.0 | 1.0~299.0 |
|---|---|---|---|---|---|---|

*TCA (total carbon aerosol) = EC+OC

*2016: Jan. 2016-Dec. 2016; 2017: Jan. 2017-Dec. 2017; 2018: Jan. 2018-Dec. 2018; 2019: Jan. 2019-Dec. 2019;

2020: Jan. 2016-Dec. 2020;

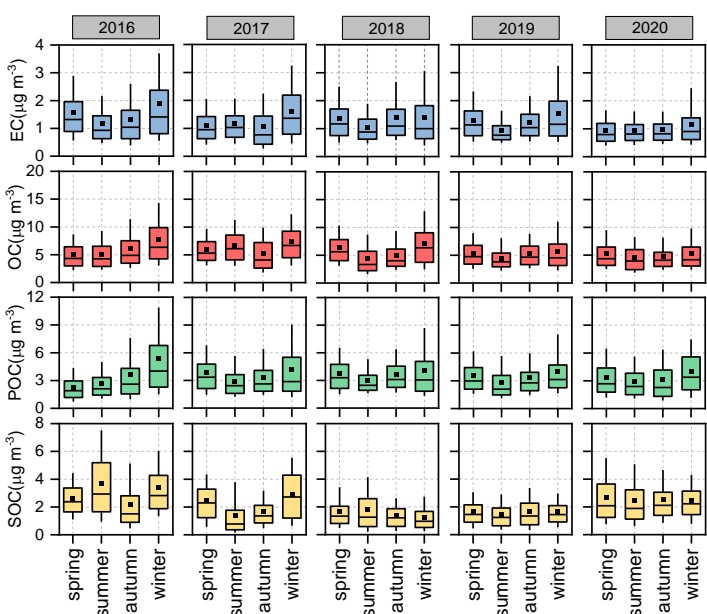

**Figure 1** Seasonal variations of carbonaceous aerosol concentrations during weekdays and weekends over different years in Dianshan Lake. (Spring: March, April, and May; summer: June, July, and August; Autumn: September, October, and November; Winter: January, February, and December). The box represents the 25th to 75th percentiles, the horizon line represents median, and the 10th and the 90th percentiles are the bottom and top whiskers, respectively.

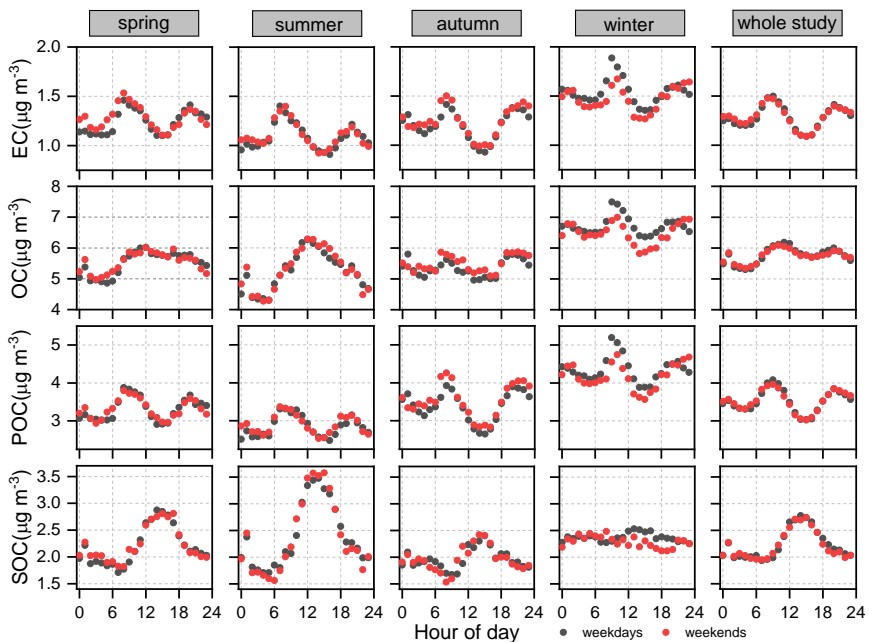

**Figure 2** Diurnal variations of carbonaceous aerosol concentrations during weekdays and weekends in four seasons and the whole study period in Dianshan Lake.

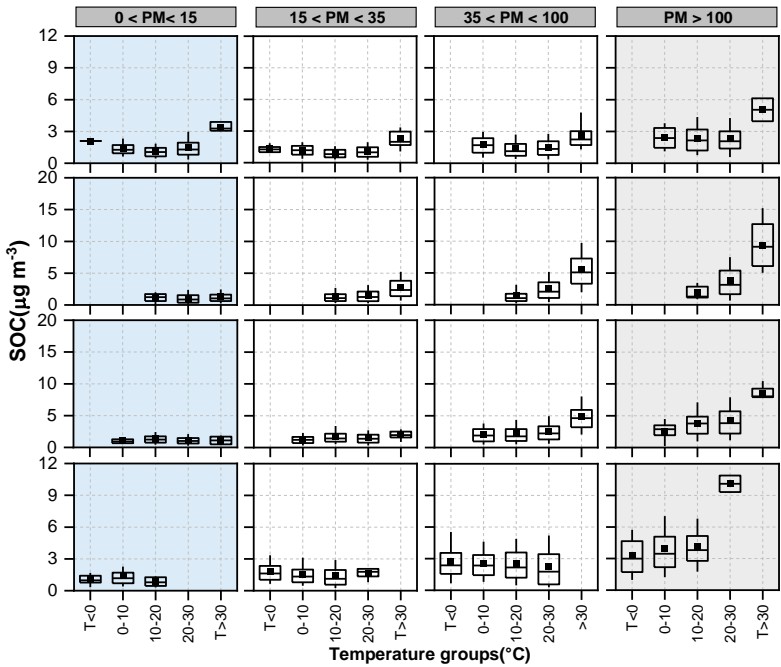

**Figure 3** The SOC dependence of temperature in four different PM$_{2.5}$ groups for each season during 2016-2020. The box represents the 25th to 75th percentiles, the horizon line represents the median, and the 10th and the 90th percentiles are the bottom and top whiskers, respectively.

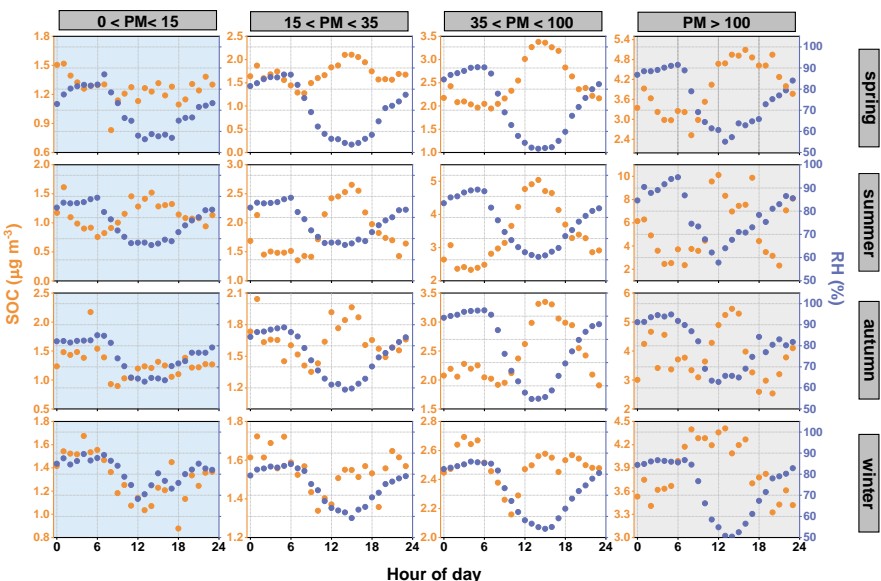

**Figure 4** Diurnal variations of SOC concentrations and RH in four different PM$_{2.5}$ groups for each season during 2016-2020.



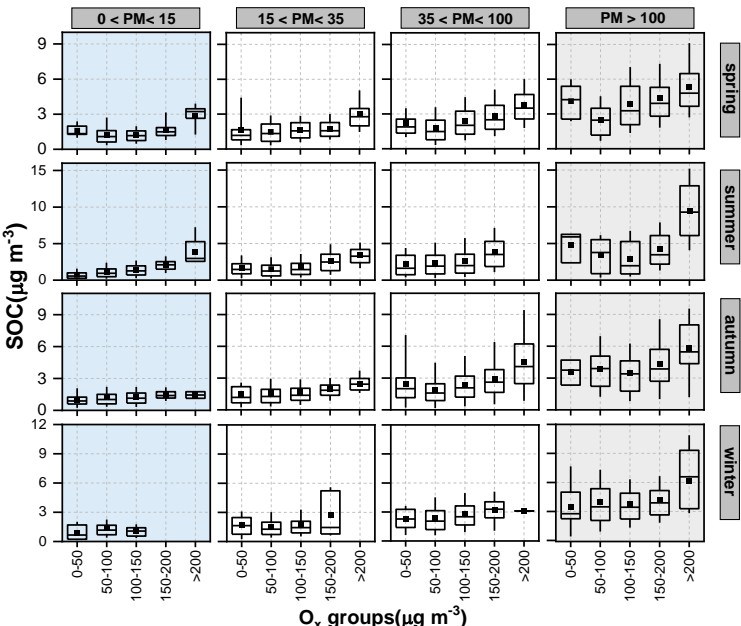

**Figure 5** The SOC dependence of $O_x$ in four different $PM_{2.5}$ groups for each season during 2016-2020. The box represents the 25th to 75th percentiles, the horizon line represents the median, and the 10th and the 90th percentiles are the bottom and top whiskers, respectively.





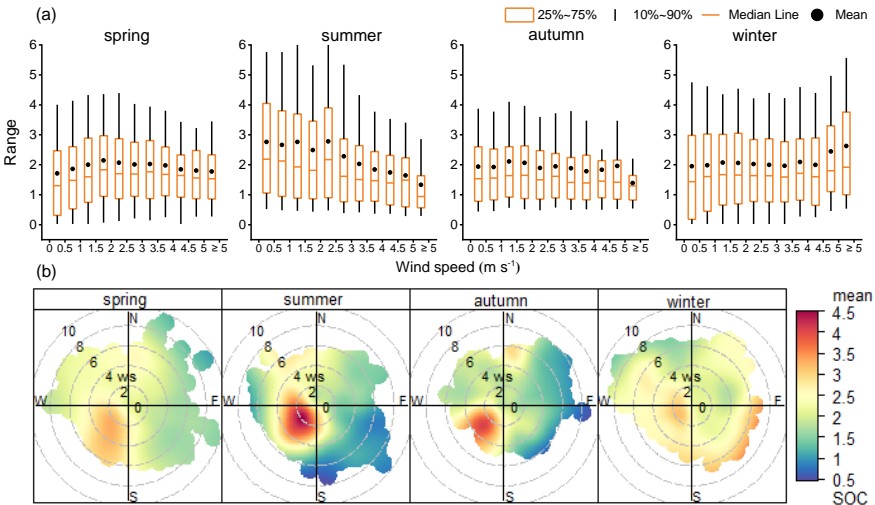

**Figure 6** (a) Box plots of SOC mass concentrations as a function of wind speed sectors over the entire sampling period; (b) Bivariate polar plots of seasonal SOC concentrations (μg m$^{-3}$) over the entire sampling period in Dianshan Lake.

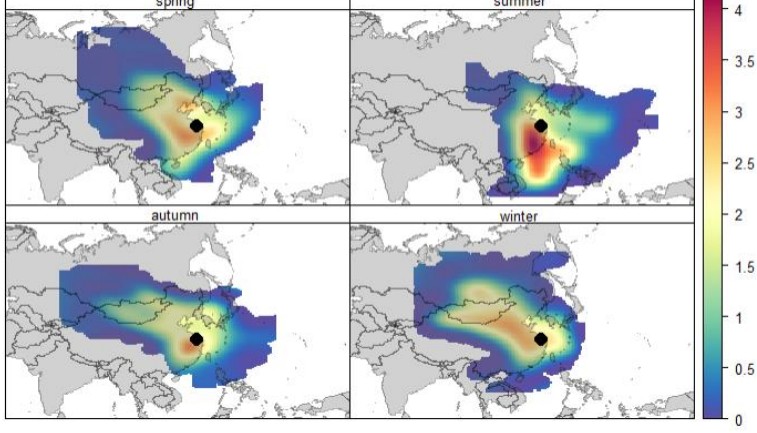

**Figure 7** Back trajectory concentrations showing mean SOC concentrations (μg m$^{-3}$) based on the CWT approach in four seasons.





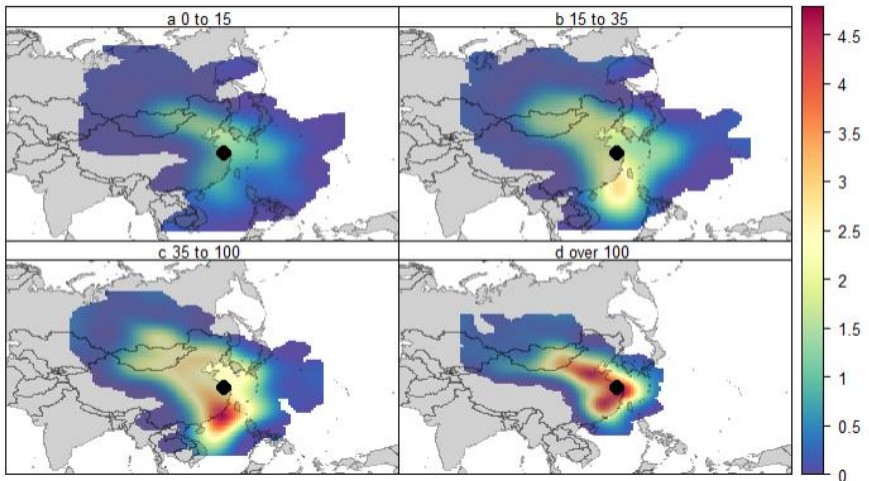

**Figure 8** Back trajectory concentrations showing mean SOC concentrations ($\mu g\ m^{-3}$) based on the CWT approach in 4 different PM groups.