# Peer review of "Measurement report: characterization and sources of the secondary organic carbon in a Chinese"

_Atmospheric Chemistry and Physics, 2022_

## Author Comment (AC1)

The authors thank the editor and anonymous referees for reviewing our manuscript, and particularly providing valuable comments and suggestions. Our responses in form of point-by-point are given.

Wang et al. reported a five-year data containing organic carbon and elementary carbon in the Shanghai, one of the biggest cities in the world. The measurement of such long-term is very valuable for policy evaluation and numerical modelling. The manuscript is well written although subjected to a few linguistic typos. The data were presented in a concise but detailed way and the main conclusion from the manuscript is clear that secondary organic aerosol (SOA) is becoming more and more important for mitigating air pollution. I believe the manuscript is suitable for publication in ACP from addressing the following issues, most of them are minor:
Response: We thank the referee for the positive comment.

The "ambient" in the title can be removed.
Response: Revised, the title now reads, "Measurement report: characterization and sources of the secondary organic carbon in a Chinese megacity over five years from 2016 to 2020".

The introduction is a bit lengthy; I understand the carbon aerosol is a large topic and there are many existing knowledge, but you don't need to introduce them all, e.g. what's the point of citing all the carbonaceous aerosol studies from line 63 to line 67.
Response: To make the introduction concise, we have now removed the detailed discussion on the OC study by Cao et al., (2007).

Line 48. You mention carbonate carbon only once, what's the point using an abbreviation
Response: Remove the abbreviation, it now reads "Carbonaceous components are classified experimentally into three fractions: elemental carbon (EC), carbonate carbon and organic carbon (OC)".

Line 55. Termed as
Response: Corrected.

Delete "in issues varying"
Response: Revised, it now reads "Carbonaceous aerosols are among the major constituents of atmospheric aerosols and their quantification is necessary for understanding the role of aerosols from the regional visibility degradation to health effects and global climate change".

Line 68 what is the relationship between this nationwide measurement with your study?
Response: It is now removed.

Line 73 you don't need a reference to introduce Shanghai?
Response: We add some description about Shanghai in Sec.1, Line 73, it now reads, "Shanghai is one of the megacities with the most rapid economic and social development in the Yangtze River Delta (YRD), China (Lin et al., 2014). The area of Shanghai is 6340.5 km$^2$ (Wang et al., 2022), and the permanent resident population of 24.89 million in 2021 (http://www.stats.gov.cn/). Along with…."

Line 96 in PM2.5, not fraction.

Response: Deleted, it now reads "The purpose of this study is to improve the understanding of the variation and sources of SOC in the PM$_{2.5}$."

Section 2 in Figure S1, please add a scale on the map.

Response: The scale on the map was added, we have updated Figure S1 (see below).

[Figure]

Section 3 as far as I understand, the concentrations is not normally distributed, the data should be reported in median (and 25th to 75th ranges)

Response: Yes, we agree that the concentrations is not normally distributed so we reported both average and median concentration in Table 1 (see below). And the average concentration here is for comparison with other references reports. We tend to report the ranges (minimum to maximum) of pollutants.

Table 1 Averages (± one standard deviation), medians, and ranges (minimum to maximum) for the OC, EC, POC, SOC and PM$_{2.5}$ concentrations (in µg m$^{-3}$) from Jan. 2016 to Dec. 2020.

|  |  | **2016** | **2017** | **2018** | **2019** | **2020** | **Whole study** |
|---|---|---|---|---|---|---|---|
| **EC** | Average | 1.50±1.17 | 1.23±0.88 | 1.31±0.88 | 1.31±0.89 | 1.00±0.64 | 1.28±0.95 |
|  | Median | 1.18 | 1.01 | 1.04 | 1.07 | 0.82 | 1.01 |
|  | Range | 0.07~11.57 | 0.01~6.27 | 0.01~9.07 | 0.08~6.85 | 0.14~5.46 | 0.01~11.57 |
| **OC** | Average | 6.03±4.01 | 6.32±3.52 | 5.79±3.58 | 5.40±3.16 | 4.99±2.93 | 5.75±3.53 |
|  | Median | 4.93 | 5.61 | 4.87 | 4.53 | 4.15 | 4.83 |
|  | Range | 0.77~41.85 | 0.41~29.49 | 0.78~29.77 | 0.78~25.96 | 0.57~26.40 | 0.41~41.85 |
| **POC** | Average | 3.48±3.23 | 3.34±2.40 | 3.61±2.67 | 3.76±2.55 | 3.45±2.27 | 3.52±2.67 |
|  | Median | 2.48 | 2.72 | 2.81 | 3.06 | 2.83 | 2.77 |
|  | Range | 0.13~37.14 | 0.02~19.41 | 0.03~22.55 | 0.19~20.71 | 0.42~17.05 | 0.02~37.14 |
|  | Average | 2.56±1.94 | 2.98±2.25 | 2.17±1.75 | 1.64±1.20 | 1.53±1.35 | 2.24±1.87 |

| | | | | | | | |
|---|---|---|---|---|---|---|---|
| **SOC** | Median | 2.10 | 2.38 | 1.71 | 1.41 | 1.20 | 1.76 |
| | Range | 0.01~18.13 | 0.01~25.79 | 0.01~19.87 | 0.01~18.84 | 0.01~14.87 | 0.01~25.79 |
| | Average | 7.53±5.06 | 7.55±4.29 | 7.10±4.38 | 6.72±3.98 | 5.98±3.50 | 7.03±4.36 |
| **TCA** | Median | 6.10 | 6.66 | 5.98 | 5.64 | 4.99 | 5.88 |
| | Range | 0.94~53.42 | 0.44~31.91 | 1.07~34.65 | 0.96~31.74 | 0.83~30.20 | 0.44~53.42 |
| | Average | 53.0±36.16 | 44.9±31.48 | 45.16±34.22 | 48.18±32.82 | 40.14±28.96 | 46.50±33.25 |
| **PM$_{2.5}$** | Median | 43.0 | 37.0 | 35.0 | 38.0 | 31.0 | 37.0 |
| | Range | 1.0~219.0 | 1.0~299.0 | 1.0~258.0 | 1.0~220.0 | 1.0~236.0 | 1.0~299.0 |

Line 172 what is the possible cause of the abrupt change of POC in 2020?

Response: The possible cause of the abrupt changes in POC in 2020 is likely associated with the Covid-19 lockdown. The effect of Covid-19 will be analysed in detail in our future study.

It now reads, "However, it dropped sharply in 2020 likely due to lockdown caused by the Covid-19 (Jia et al., 2020)"

Line 213 why is the weekend -weekday pattern important?

Response: Studies on weekend-weekday patterns can provide insights into the variations of traffic emissions since traffic volume are expected to be larger on weekdays than weekends especially in urban area. However, we found that this pattern is not held true for highway traffic. In particular, according to a previous literature report (Chang et al., 2017), the observational data from 2010 to 2014 showed that the concentration of EC on working days was greater than that on weekends because the traffic volume was significantly higher on weekdays than on weekends. However, Shanghai has officially implemented a traffic restriction system in 2016. In this study, the sampling site is located near tourist attractions and is not in the traffic restricted area of Shanghai, which is near the national expressway entering and leaving Shanghai (the straight-line distance is no more than two kilometers). It is speculated that the heavy traffic flow due to the attraction of the nearby tourist sites during spring and autumn weekends may lead to high EC emissions.

The manuscript was focused on secondary organic aerosol, but a large fraction in 3.1 was given to primary organic aerosol.

Response: We agree that we focused on SOC. Section 3.1 gave an overview of the variation of carbonaceous aerosol (SOC + POC +EC).

Section 3.2.1 It is intriguing to see the SOC increased with temperature, the authors represent a detailed analysis on the effect of temperature on SOC concentrations, but it is unclear how temperature impacts SOC. I understand that temperature can boost biogenic VOC in boreal area that act as SOC precursors, but in mega city in Shanghai it is uncommon. A possible reason should at least be given.

Response: We agree that high temperature can boost biogenic VOC emissions. However, high temperatures are also usually associated with high solar radiation intensities (Shrestha et al., 2019), which can promote the photochemical oxidation of both biogenic and anthropogenic VOC, increasing the SOC concentrations as observed in this study. We have now added the discussion on the possible reason of the relationship

between SOC and temperature.

It now reads, "High temperatures can boost the emission of SOC precursors, e.g., biogenic VOCs as well as anthropogenic VOCs from e.g., solvent use (Zheng et al., 2018). Moreover, high temperatures are usually associated with a strong solar radiation (Shrestha et al., 2019), which can promote the photochemical oxidation of both biogenic and anthropogenic VOCs, increasing SOC concentrations."

Line 311. "The concentration of Ox"

Response: Added. It now reads "The concentration of oxidant $O_x$ ($O_x = O_3 + NO_2$) is usually used as a proxy to indicate the atmospheric oxidizing capacity associated with photochemical reactions (Wang et al., 2017)."

Line 392. Does this suggest that SOC observed in Shanghai is mainly originated from regional transport?

Response: Yes. High winds are found to be associated with SOC, suggesting regional transport rather than local promotion. The CWT plots shows the potential regional sources of SOC.

Chang Y, Deng C, Cao F, Cao C, Zou Z, Liu S, et al. Assessment of carbonaceous aerosols in Shanghai, China – Part 1: long-term evolution, seasonal variations, and meteorological effects. Atmos. Chem. Phys. 2017; 17: 9945-9964.

Jia H, Huo J, Fu Q, Duan Y, Lin Y, Jin X, et al. Insights into chemical composition, abatement mechanisms and regional transport of atmospheric pollutants in the Yangtze River Delta region, China during the COVID-19 outbreak control period. Environmental Pollution 2020; 267: 115612.

Shrestha AK, Thapa A, Gautam H. Solar Radiation, Air Temperature, Relative Humidity, and Dew Point Study: Damak, Jhapa, Nepal. International Journal of Photoenergy 2019; 2019: 8369231.

Wang S, Zhao Y, Han Y, Li R, Fu H, Gao S, et al. Spatiotemporal variation, source and secondary transformation potential of volatile organic compounds (VOCs) during the winter days in Shanghai, China. Atmospheric Environment 2022; 286: 119203.

Zheng B, Tong D, Li M, Liu F, Hong C, Geng G, et al. Trends in China's anthropogenic emissions since 2010 as the consequence of clean air actions. Atmos. Chem. Phys. 2018; 18: 14095-14111.

---

## Author Comment (AC2)

The authors thank the editor and anonymous referees for reviewing our manuscript, and particularly providing valuable comments and suggestions. Our responses in form of point-by-point are given.

The manuscript entitled "Measurement report: characterization and sources of the ambient secondary organic carbon in a Chinese megacity over five years from 2016 to 2020" conducted a long-term field campaign at a regional site in the YRD region from 2016 to 2020 and aimed to investigate the characteristics of carbonous aerosol pollution and their seasonal and diurnal variations, as well as the relationship between the meteorological factors and carbonaceous aerosol concentrations. This study enhanced the understanding of the variation and sources of SOC in the PM2.5 fraction, and was in favor of evaluation of the effectiveness of the current air pollution control policies. The manuscript is overall well organized, and can be read easily. I broadly agree with the discussions and findings of this manuscript. I therefore recommend a minor revision of this manuscript before final publication in ACP.

Response: We thank the referee for the positive comment.

In the conclusion, it is difficult for me to find research findings with strong regularity or regional characteristics of the Yangtze River Delta. Therefore, it is suggested to condense the conclusion.

Response: We have now condensed the conclusion, it now reads, "…Our results elucidate the trends in SOC and POC over recent years in Shanghai, which are important for evaluating the effectiveness of the air pollution control measures and holding important implications for policymaking. Given that SOC was associated with high temperature and regional transport, global warming is likely increasing the importance of SOC. Since SOC is regional, combined efforts in reducing regional sources of SOC precursors are needed to further reduce the air pollution events in Shanghai."

In 3.2.3, The discussion on the formation of photochemistry should have become one of the highlights of the paper, but unfortunately, the reviewer found that the author basically stayed at the level of the discussion on the correlation between ox and SOC, and lacked in-depth analysis of radiation intensity and liquid phase processes, suggesting further in-depth discussion.

Response: We used the concentration of oxidant of $O_x$ ($O_3+NO_2$) as a proxy for the atmospheric oxidizing capacity associated with photochemical reactions. The positive relationship between SOC and Ox is suggesting a photochemical formation pathway of SOC (see Fig. 5). The radiation intensity is usually stronger in summer than in winter. In this study, we also observed a concurrent increase in SOC with Ox during winter pollution periods (PM>100 μg m$^{-3}$; Fig. 4), suggesting the importance of photochemical reactions even in winter. Due to the lack of in-situ measurement of radiation intensity, further discussion of the impact of radiation is not included.

The liquid phase of SOC formation is often reported during winter Haze in north China with a positive relationship between SOC and RH (Lin et al., 2020; An et al., 2019; Sun et al., 2015; Chen et al., 2019). In this study, we found RH is negatively correlated with SOC. Therefore, we concluded liquid phase chemistry was likely less important compared to photochemical formation. Because of this, we tended not to expand our discussion on liquid phase chemistry.